environmental chemistry

lignin model compounds, chlorine dioxide, adsorbable organic halogens, bleaching

**Author for correspondence:**
Shuangquan Yao
e-mail: yaoshuangquan@gxu.edu.cn

This article has been edited by the Royal Society of Chemistry, including the commissioning, peer review process and editorial aspects up to the point of acceptance.

# Effect of lignin structure on adsorbable organic halogens formation in chlorine dioxide bleaching

Lisheng Shi[1,2], Jiayan Ge[1,2], Shuangxi Nie[1,2], Chengrong Qin[1,2] and Shuangquan Yao[1,2]

[1]College of Light Industrial and Food Engineering, Guangxi University, Nanning 530004, People's Republic of China
[2]Guangxi Key Laboratory of Clean Pulp and Papermaking and Pollution Control, Nanning 530004, People's Republic of China

iD SY, 0000-0003-4982-998X

Adsorbable organic halogens (AOX) are formed in pulp bleaching as a result of the reaction of residual lignin with chlorine dioxide. The natural structure of lignin is very complex and it tends to be damaged by various extraction methods. All the factors can affect the study about the mechanism of AOX formation in the reaction of lignin with chlorine dioxide. Lignin model compounds, with certain structures, can be used to study the role of different lignin structures on AOX formation. The effect of lignin structure on AOX formation was determined by reacting phenolic and non-phenolic lignin model compound with a chlorine dioxide solution. Vanillyl alcohol (VA) and veratryl alcohol (VE) were selected for the phenolic and non-phenolic lignin model compound, respectively. The pattern consumption of lignin model compounds suggests that both VA and VE began reacting with chlorine dioxide within 10 min and then gradually steadied. The volume of AOX produced by VE was significantly higher than that produced by VA for a given initial lignin model compound concentration. In a solution containing a combination of VA and VE in chlorine dioxide, VE was the dominant producer of AOX. This result indicates that the non-phenolic lignin structure was more easily chlorinated, while the phenolic lignin structure was mainly oxidized. In addition, AOX content produced in the combined experiments exceeded the total content of the two separate experiments. It suggested that the combination of phenolic and non-phenolic lignin structure can promote AOX formation.

# 1. Introduction

In the pulp bleaching process, residual lignin reacts with chlorine dioxide to produce adsorbable organic halogens (AOX) [1–4]. AOX has a degree of toxicity, causing genetic and reproductive damage in both aquatic and terrestrial animals, including humans [5–7]. Lignin has a very complex structure because its monomers are connected by many different covalent bond types [8,9]. Furthermore, the lignin–carbohydrate complexes network increases the complexity of lignin research in many circumstances [10,11]. Because the structure of lignin changes easily during the extraction and separation processes [12,13], it is difficult to use natural lignin directly to accurately study its reactivity with other chemical structures [14,15].

The simplest methods to research these structural changes employ lignin model compounds [16–18]. Many lignin model compounds have been synthesized and applied in various studies [19–21]. Monomeric and dimeric lignin model compounds with relatively simple structure are common, due to the difficulty of polymer synthesis [22,23]. Many studies have focused on the degradation of lignin model compounds. Nie *et al.* [18] studied the oxidation kinetics of non-phenolic lignin model compound 1-(3,4-dimethoxyphenyl) ethanol (MVA) to provide a reference for lignin degradation in the chlorine dioxide stage of elemental chlorine-free (ECF) bleaching. Ni *et al.* showed that the degradation of phenolic and non-phenolic lignin model compounds contains three independent parallel reactions. Hamzeh *et al.* [24] indicated that the primary lignin oxidation products could be the cause of overconsumption of oxidant in a chlorine dioxide bleaching stage. The structural changes of certain lignin compounds during the reaction process have also been studied [25,26]. However, the present understanding of the effect of the combination of different structures on AOX formation during bleaching is inadequate [27]. Theoretically, reaction rates and sequences vary between the different lignin structures. In turn, the performance of a solution containing many structures should differ from that of a solution containing a single structure [28–30]. It is unclear how the combination of various structures influences a reaction. This issue may be studied using a set of lignin model compounds with carefully selected structures.

In this research, the effect of lignin structures on AOX formation was studied using phenolic and non-phenolic lignin model compound. 4-Hydroxy-3-methoxybenzyl alcohol (vanillyl alcohol) and 3,4-dimethoxybenzyl alcohol (veratryl alcohol) were selected as the phenolic and non-phenolic lignin model compound, respectively. The effect of lignin structures on AOX formation can be studied easily using model compounds but would be difficult to elucidate using the natural lignin macromolecule. The aim of the present work was to assess the effect of the combination of different lignin structures on AOX formation by the reaction of phenolic and non-phenolic lignin model compound with chlorine dioxide in solution. These results provide insight into the mechanism of AOX formation in the reaction of lignin with chlorine dioxide in pulp bleaching.

# 2. Material and methods

## 2.1. Materials

Vanillyl alcohol (VA) and veratryl alcohol (VE) were purchased from Aladdin (Shanghai, China). Activated carbon and ceramic cotton were purchased from Analytic-Jena Instrument Company (Jena, Germany). Chlorine dioxide solution was procured from a paper mill located in Guangxi, China, and had an effective chlorine concentration of $18.5\,\mathrm{g\,l^{-1}}$. All other reagents were analytical grade and purchased from Aladdin (Shanghai, China).

## 2.2. Experiments

Chlorine dioxide solution was transferred into a conical bottle and adjusted to pH 3 with sulfuric acid. Then, the respective lignin model compound was transferred to the chlorine dioxide solution and reacted at 70°C with magnetic stirring at 150 r.p.m. The reaction was terminated by the oxidation of sodium sulfite with residual chlorine dioxide. In addition, the reaction mixture was diluted immediately in deionized water to achieve reaction termination. The reaction mixture was stored in a refrigerator and analysed as soon as possible.

## 2.3. AOX analysis

The AOX content of the reacting solution was measured using a MultiX2500 AOX analyser (Jena, Germany). The general methods were as follows: the pH of the reaction solution was adjusted to 2 with nitric acid. Then, 100 ml of the diluted solution and 50 mg of activated carbon were transferred into a conical bottle and shaken for 60 min in a thermostatic oscillator at a speed of 150 r.p.m. at room temperature (25°C). Quartz filter cups were used to collect the activated carbon, and the adsorbed inorganic chloride was washed with sodium nitrate. The water was drained, and the quartz filter cup and activated carbon were then combusted in a furnace. The AOX content was quantified by the microcoulomb titration method [31,32].

## 2.4. HPLC analysis

The mass of the lignin model compounds was measured using high-performance liquid chromatography (Agilent 1260 Infinity, USA). Both model compounds (phenolic or non-phenolic) were analysed on a Diamonsil C18 column (150 × 4.6 mm, 5 µm) using a mobile phase composed of acetonitrile–water (10/90) (vol) and 0.1% acetic acid, a flow rate of 0.3 ml min$^{-1}$, a temperature of 50°C and UV detection at 280 nm [24].

## 2.5. GC–MS analysis

The chemical composition of the reaction solution was quantitatively determined by GC–MS (Agilent 7890A, USA) with a VF-1701 MS column (30 m × 250 µm × 0.25 µm). The basic method and process were described by Yao and co-workers [31].

# 3. Results and discussion

## 3.1. Effect of reaction time on AOX formation

The AOX content and consumption of lignin model compounds were measured over the course of the reaction to compare the reaction of chlorine dioxide solution with phenolic and non-phenolic lignin model compounds. Each individual reaction was performed in a solution with an initial lignin model compound of 6.2 mmol and concentration of 124 mmol l$^{-1}$. Sampling analysis was performed at 2, 4, 6, 8, 10, 15, 20, 30, 40, 50 and 60 min of reaction time.

Figure 1 shows that the AOX formation occurs quickly. AOX were produced rapidly at the beginning of the reaction of VA with chlorine dioxide solution; an upward trend of AOX content was visible within 10 min. The AOX content increased from 201.4 mg l$^{-1}$ at 2 min to 232 mg l$^{-1}$ at 10 min and then steadied at increased reaction times. One major reason for this trend is the rapid consumption of VA (figure 2): although 5.5 mmol of VA was consumed within 10 min, accounting for 88.7% of the initial content, the total consumption after 60 min was only 6.0 mmol. This is in agreement with the conclusions of Pardeep *et al*. [33], who found that the oxidation of phenol by chlorine dioxide in water solutions is a very fast step. In addition, AOX are mainly produced by the reaction of hypochlorite and chlorine with lignin. Because of the acidic reaction conditions, chlorine dioxide is converted to hypochlorite, which is then converted into chlorine at pH values below 2. Therefore, there is more hypochlorite and chlorine in the reaction system at the beginning of the reaction, which could react with the lignin model compounds to produce a large quantity of AOX quickly. However, the rate of AOX formation rate slows as the hypochlorite and chlorine are gradually consumed.

AOX was also formed rapidly at the beginning of the reaction when VE reacted with chlorine dioxide, reaching 1036 mg l$^{-1}$ after 10 min. Then, the AOX content gradually stabilized due to the consumption of VE and hypochlorite and chlorine in the reaction system. The actual consumption of VE after 60 min was only 3.3 mmol, which was significantly lower than that of VA. This is mainly attributed to the high reactivity of the phenolic hydroxyl of VA with chlorine dioxide. Under the condition of excessive lignin model, less VE was consumed by quantitative chlorine dioxide. This is due to the low reactivity of non-phenolic lignin. The complete reaction of VE requires more chlorine dioxide than VA [34,35].

In addition, at the same initial concentration, significantly more AOX was produced with VE than with VA. VA containing phenolic structure has higher reactivity. Phenolic hydroxyl groups were converted into free radicals, for it was easily attacked by chlorine dioxide, and then the oxidation fracture of the benzene ring. Most free radicals are further oxidized by chlorine dioxide to form

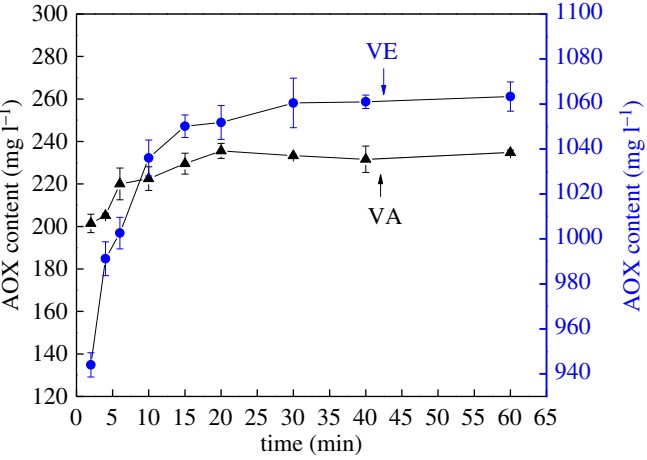

**Figure 1.** Comparison of the effect of VA and VE on AOX formation (VA is phenolic and VE is non-phenolic).

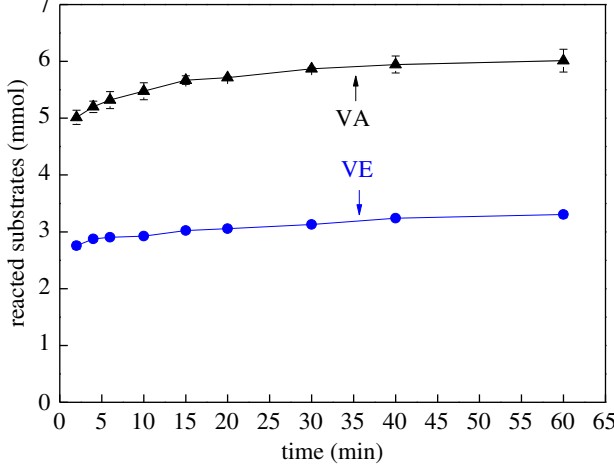

**Figure 2.** Comparison of the consumption of VA and VE (VA is phenolic and VE is non-phenolic).

chlorolipids. It was further decomposed into muconic acid, ortho-quinone, para-quinone and oxacyclopropane structures for its unstable structure. Oxidative fracture of aromatic structures was the main reaction of VA in chlorine dioxide solution. Therefore, the lower content of AOX was generated from VA. VE has lower reactivity due to the absence of phenolic hydroxyl group. The hydroxymethyl side chain was easily oxidized by chlorine dioxide. Then chlorine substitution occurred on the aromatic ring. More content of AOX was generated at a lower reaction rate.

Phenolic and non-phenolic lignin model compounds were reacted simultaneously to study the effect of mixed solutions on AOX content and lignin model compound consumption. VA and VE had an initial content of 6.2 mmol and initial concentration of 62 mmol $l^{-1}$. Thus, the total initial concentration of lignin model compounds was 124 mmol $l^{-1}$. Sampling analysis was performed at 2, 4, 6, 8, 10, 15, 20, 30, 40, 50 and 60 min of reaction time.

The curve of AOX formation is similar to that of the separated experiments despite the combination of VA and VE in the chlorine dioxide solution (figure 3). The AOX content increased within 10 min and ultimately reached 1224 mg $l^{-1}$. These results indicated that the AOX formation by VA and VE with chlorine dioxide solution was higher than the total AOX content formed in the two separate experiments. The actual consumption of VA and VE was 6.19 and 5.02 mmol, respectively, which is almost double their consumption when reacted separately.

## 3.2. Concentration of lignin model compounds

The effects of initial concentration of lignin model compound on AOX formation in the separate reactions of phenolic and non-phenolic lignin models were compared (figure 4). Initial VA and VE concentrations

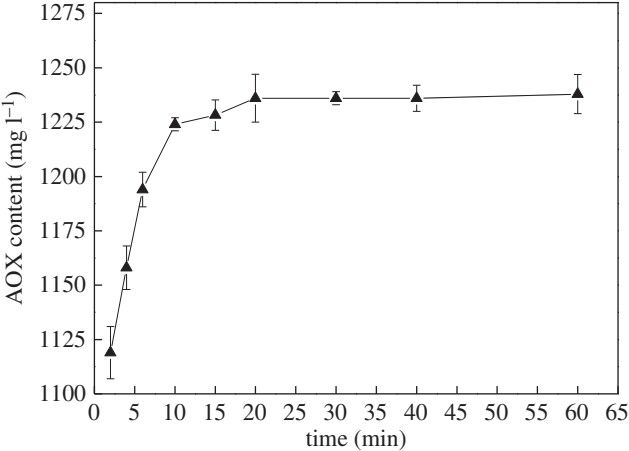

**Figure 3.** Effect of combination of VA and VE on AOX formation.

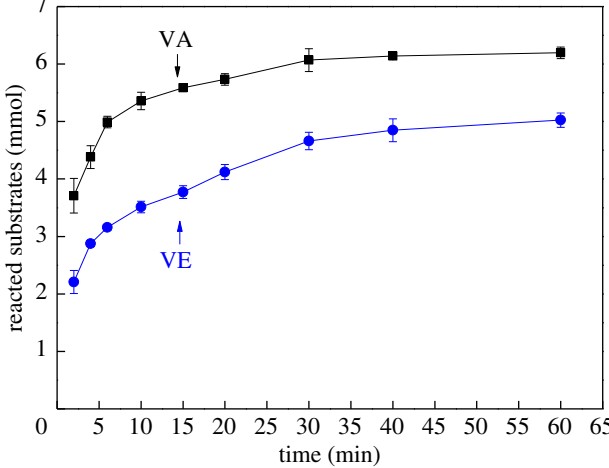

**Figure 4.** Effect of combination of VA and VE on lignin model compounds consumption (VA is phenolic and VE is non-phenolic).

of 12.4, 62, 124, 186 and 248 mmol l$^{-1}$ were tested. The AOX content was analysed after 60 min of reaction under simulated bleaching conditions.

Figure 5 illustrates the increase in AOX content with the concentration of lignin model compound. The curve of AOX formation shows a significant increase with the concentration below 124 mmol l$^{-1}$ and gradual stabilization thereafter. It shows that the promotion effect of concentration on AOX formation decreases significantly once the concentration of lignin model compounds exceeds 124 mmol l$^{-1}$. The lignin model compounds can be oxidized completely by chlorine dioxide solution because of their low concentration. Therefore, the AOX content increases rapidly with concentration of the model compounds in a certain range. Above this concentration range, its effect on AOX formation is less pronounced.

The AOX content formed by the reaction of VE with chlorine dioxide solution was 60.9 mg l$^{-1}$ creating a total concentration of 12.4 mmol l$^{-1}$, which is slightly higher than that formed by VA with chlorine dioxide solution (55.4 mg l$^{-1}$). VA containing phenolic structure has higher reactivity. Phenolic hydroxyl groups were converted into free radicals, for it was easily attacked by chlorine dioxide and then the oxidation fracture of the benzene ring. Most free radicals are further oxidized by chlorine dioxide to form chlorolipids. It was further decomposed into muconic acid, ortho-quinone, para-quinone and oxacyclopropane structures for its unstable structure. Oxidative fracture of aromatic structures was the main reaction of VA in chlorine dioxide solution. Therefore, the lower content of AOX was generated from VA. VE has lower reactivity due to the absence of phenolic hydroxyl group. The hydroxymethyl side chain was easily oxidized by chlorine dioxide. Then chlorine substitution occurred on the aromatic ring. More content of AOX was generated at a lower reaction rate. When the concentration is 248 mmol l$^{-1}$, the AOX content formed by the reaction of VE with chlorine dioxide

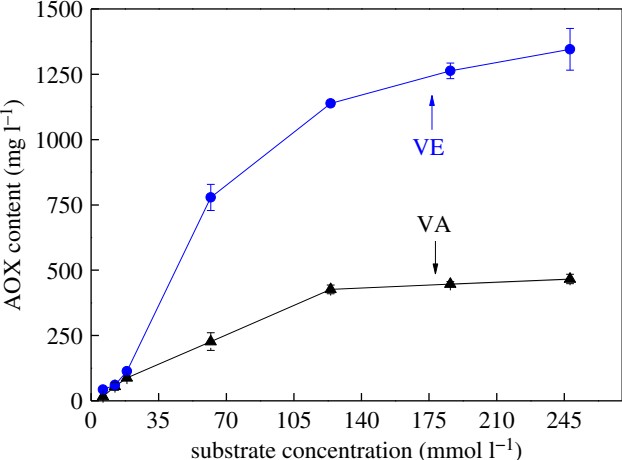

**Figure 5.** Comparison of the effect of initial concentration of VA and VE on AOX formation (VA is phenolic and VE is non-phenolic).

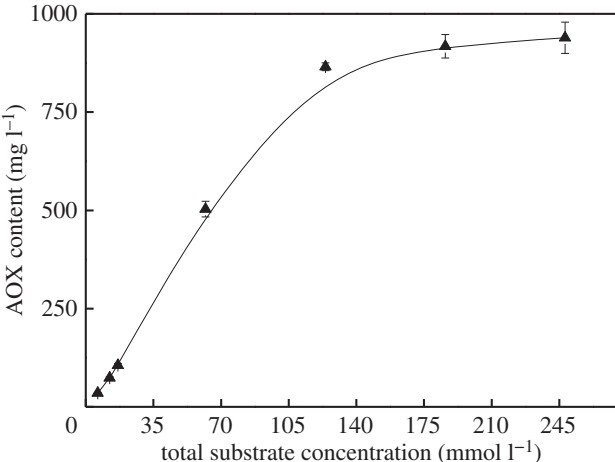

**Figure 6.** Effect of total concentration of VA and VE on AOX formation.

solution was 1345.6 mg l$^{-1}$ and almost triple the quantity formed by VA with chlorine dioxide solution (466.4 mg l$^{-1}$). Unsurprisingly, at identical initial model compound concentrations, the AOX content produced by the reaction of VE with chlorine dioxide solution was higher than that by the reaction of VA with chlorine dioxide solution. In addition, the gap in AOX formation between the two separate reactions widened with an increase in concentration, which indicates that the reaction of VE with chlorine dioxide is more easily influenced by concentration.

The effect of the total concentration of VA and VE on AOX formation in the combined reaction was studied. The combined initial concentrations of lignin model compounds were 12.4, 62, 124, 186 and 248 mmol l$^{-1}$, and the molar ratio of VA and VE was 1 : 1.

Figure 6 shows that the relationship between AOX formation and the total concentration of lignin model compounds is similar to that seen in the separate reactions. The AOX content is 865.6 mg l$^{-1}$ for a total concentration of 124 mmol l$^{-1}$, which is 10-fold higher than that at a total concentration of 12.4 mmol l$^{-1}$ (73.6 mg l$^{-1}$). This suggests that the effect of total concentration on AOX formation is larger at concentrations below 124 mmol l$^{-1}$. However, the AOX content cannot increase significantly any more once the total concentration exceeds 124 mmol l$^{-1}$.

The comparison of the AOX content of separate and combined experiments of VA and VE with chlorine dioxide solution is presented in table 1. The total AOX content of the separate reaction is less than that of combined reaction of VA and VE with chlorine dioxide solution at the same total concentration. Under the condition of excessive lignin model, less VE was consumed by quantitative chlorine dioxide. This is due to the low reactivity of non-phenolic lignin. The complete reaction of VE requires more chlorine dioxide than VA. In combination reaction of VA and VE, the dosage of chlorine dioxide reacting with VE increased. The consumption of VE increased compared to that in separated

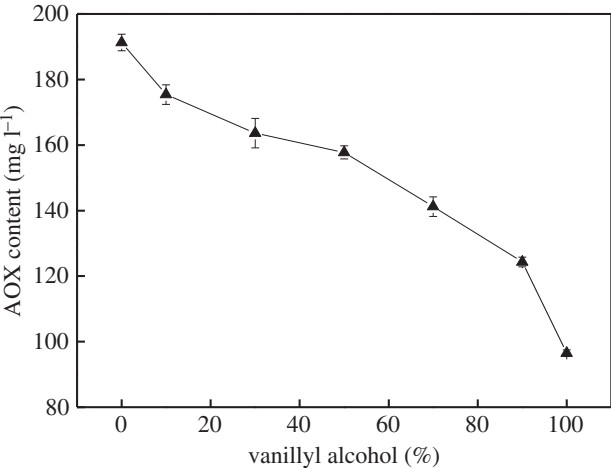

**Figure 7.** Effect of the molar ratio of VA and VE on AOX formation.

**Table 1.** Comparison of the AOX formation of separated and combined reaction of VA and VE.

| total concentration (mmol l$^{-1}$) | AOX content of separated reaction | | | AOX content of combined reaction (mg) |
| --- | --- | --- | --- | --- |
| | VA (mg l$^{-1}$) | VE (mg l$^{-1}$) | total (mg) | |
| 6.20 | 16.81 | 43.10 | 59.91 | 70.16 |
| 12.40 | 55.73 | 60.90 | 116.63 | 147.28 |
| 18.60 | 87.69 | 113.21 | 200.90 | 211.64 |
| 62.00 | 226.70 | 778.19 | 1004.89 | 1006.40 |
| 124.00 | 426.43 | 1138.77 | 1565.20 | 1731.22 |
| 186.00 | 446.38 | 1263.20 | 1709.58 | 1835.26 |
| 248 | 466.41 | 1345.61 | 1812.02 | 1878.40 |

reaction of VE. Meanwhile, the VA shared similar consumption in separated and combined reaction. It indicates that the combination of VA and VE has a synergistic effect on the formation of AOX in the reaction with chlorine dioxide solution.

## 3.3. Molar ratio of lignin model compounds

The effect of the molar ratio of phenolic and non-phenolic lignin model compounds on AOX formation was studied. The VA and VE had a total content of 3.1 mmol and concentration of 62 mmol l$^{-1}$. The molar ratio of VA and VE was 0/1, 0.1/0.9, 0.3/0.7, 0.5/0.5, 0.7/0.3, 0.9/0.1 and 1/0, respectively. The AOX formation was analysed after reacting for 60 min under the condition of bleaching.

Figure 7 shows that AOX content decreases as the proportion of VA increases when the total concentration of lignin model remains constant. It indicates that VE plays a major role in the formation of AOX, due to the absence of a phenolic hydroxyl. The AOX content reaches its largest value (191.3 mg l$^{-1}$) as the proportion of VE is 100%. The AOX formation decreases slowly with an increase in VA as the proportion of VA falls below 50%, which decreases by 17.5% from 191.3 to 157.75 mg l$^{-1}$. However, the AOX formation decreases significantly with the further increases in the proportion of VA. When VA reaches 100%, it decreases by 38.8% from 157.75 to 96.5 mg l$^{-1}$. This result confirms that VE contributes more to AOX formation, and so, it is the VE content that more greatly influences AOX formation in the reaction of VA and VE with chlorine dioxide solution.

## 3.4. Identification of chlorinated products

The products of the reaction of VA with chlorine dioxide solution were separated and detected by GC–MS. The chlorinated products obtained are listed in table 2. In these experiments, 1,4-dibromide benzene

**Table 2.** Chlorinated products formed by reaction of VA with chlorine dioxide solution.

| | time (min) | content (%) | substance | molecular structure |
|---|---|---|---|---|
| 1 | 6.59 | 1 | 1,4-dibromide benzene (internal standard) | |
| 2 | 9.394 | 0.02 | 2-chloro-1,4-dimethoxybenzene | |
| 3 | 11.637 | 0.92 | 2-chlorobiphenyl | |
| 4 | 12.689 | 0.56 | 2-chloro-5-methoxy-1-hydroxynaphthalene | |
| 5 | 12.846 | 0.18 | 2-chloro-3,6-dihydroxy-benzoic acid, methyl ester | |
| 6 | 13.195 | 0.99 | 4-chlorobiphenyl | |

**Table 3.** Chlorinated products formed by reaction of VE with chlorine dioxide solution.

| | time (min) | content (%) | substance | molecular structure |
|---|---|---|---|---|
| 1 | 6.584 | 1 | 1,4-dibromide benzene | |
| 2 | 8.485 | 12.64 | 4-chloro-1,2-dimethoxybenzene | |
| 3 | 8.662 | 0.91 | 3-chloro-1,2-dimethoxybenzene | |
| 4 | 10.368 | 0.05 | 1,2-dichloro-4,5-dimethoxybenzene | |
| 5 | 12.287 | 0.55 | 2-chloroveratraldehyde | |
| 6 | 12.521 | 18.95 | 1,2-benzenedicarbonyl dichloride | |

was added as the internal standard compound. Chlorobenzenes such as 2-chlorobiphenyl and 4-chlorobiphenyl were common among these products while chlorophenols were relatively minor components.

The products of the reaction of VE with chlorine dioxide solution were separated and detected by GC−MS (table 3). The type and content of chlorinated products produced by the reaction of VE and chlorine dioxide solution are quite different from that of VA. The chlorinated products produced by the reaction of VE had a higher relative content than those produced with VA. For example, 1,2-

**Table 4.** Chlorinated products formed by combined reaction of VA and VE with chlorine dioxide solution.

| | time (min) | content (%) | substance | molecular structure |
|---|---|---|---|---|
| 1 | 6.589 | 1 | 1,4-dibromide benzene | |
| 2 | 8.484 | 9.13 | 4-chloro-1,2-dimethoxybenzene | |
| 3 | 8.664 | 0.13 | 3-chloro-1,2-dimethoxybenzene | |
| 4 | 9.39 | 0.06 | 2-chloro-1,4-dimethoxybenzene | |
| 5 | 11.696 | 0.44 | 3-chlorobiphenyl | |
| 6 | 11.946 | 0.56 | 2-chloro-3,6-dihydroxy-benzoic acid, methyl ester | |
| 7 | 12.288 | 0.38 | 2-chloroveratraldehyde | |
| 8 | 12.517 | 11.44 | 1,2-benzenedicarbonyl dichloride | |
| 9 | 12.635 | 1.44 | 5-chlorovanillic acid | |

benzenedicarbonyl and 4-chloro-1,2-dimethoxybenzene had a relative content of 18.95% and 12.64%, respectively. These results also support that the total AOX content produced by VE is significantly greater than that by VA.

The chlorinated products of the combined reaction of VA and VE with chlorine dioxide solution are shown in table 4. These included the main products found in the separate reactions. For example, 1,2-benzenedicarbonyl dichloride, which had the highest relative content of 11.44%, and the 4-chloro-1,2-dimethoxybenzene, with a content of 9.13%, are also the main products formed in the reaction of VE with chlorine dioxide solution. This indicates that the reaction of VE with chlorine dioxide solution plays a dominant role in the formation of AOX. In addition, some unique chlorinated products were formed in the combined reaction. These include 5-chlorovanillic acid, which is formed by the oxidation of hydroxyl groups of VA to carboxyl groups and the substitution of hydrogen atoms at the five positions by chlorine atoms.

# 4. Conclusion

The AOX content produced by the reaction of VE with chlorine dioxide solution is higher than that of VA with chlorine dioxide solution, although the consumption of VE is smaller than that of VA. This indicates that the phenolic lignin model compound is more likely to react with chlorine dioxide solution even though the non-phenolic lignin model compound contributed more to the formation of AOX. In addition, the amount of AOX generated in the combined reaction of VA and VE with chlorine dioxide

was higher than that in the separate reactions. This indicates that the interaction between VA, VE and the primary oxidized products tends to promote the formation of AOX.

Data accessibility. The datasets supporting this article have been uploaded as part of the manuscript and electronic supplementary material.

Authors' contributions. S.Y. designed the study. L.S., J.G., S.N. and C.Q. collected all data for analysis. L.S. and S.Y analysed the data, interpreted the results and wrote the manuscript. All authors gave final approval for publication.

Competing interests. We declare we have no competing interests.

Funding. This project was sponsored by the National Natural Science Foundation of China (31760192). This project was supported by the Guangxi Natural Science Foundation of China (2016GXNSFBA380234).

Acknowledgements. We thank Guangxi Key Laboratory of Clean Pulp and Papermaking and Pollution Control for help.

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
