## [Reviewer comments · Royal Society Open Science]

Review History

RSOS-182024.R0 (Original submission)

Review form: Reviewer 1

Is the manuscript scientifically sound in its present form?

Yes

Are the interpretations and conclusions justified by the results?

Yes

Is the language acceptable?

Yes

Is it clear how to access all supporting data?

Yes

Do you have any ethical concerns with this paper?

No

Have you any concerns about statistical analyses in this paper?

No

Recommendation?

Accept with minor revision (please list in comments)

Comments to the Author(s)

Effect of lignin structure on adsorbable organic halogens (AOX) formation was investigated by authors. Vanillyl alcohol (VA) and veratryl alcohol (VE) were selected for the phenolic and non-phenolic lignin model compounds, respectively. It has practical and scientific significance in elucidating the reaction pathway of lignin with ClO₂ and therefore reducing chlorinated pollutants. I believe the manuscript can be accepted after the following minor revisions.

2. Introduction. "The chemical structure of lignin is complex, and the lignin macromolecules have a great variety of bonds". Why were phenolic lignin model compound and non-phenolic lignin model compound selected to study?

3.2. Experiments. "Then, the respective lignin model compound was transferred to the chlorine dioxide solution and reacted at 70 °C with magnetic stirring". How many rpm? Please explain the experiment more clearly.

4.1. Effect of reaction time on AOX formation. "This is in agreement with the conclusions of C. Bogatu, who found that the oxidation of phenolic compounds with chlorine dioxide in water solutions contains a very fast step". Please insert references.

4.1. Effect of reaction time on AOX formation. "AOX was also formed rapidly at the beginning of the reaction when VE reacted with chlorine dioxide, reaching 1036 mg•L⁻¹ after 10 minutes". Please revise the time unit of the manuscript.

5. Conclusion. "The AOX content produced by the reaction of VE with chlorine dioxide solution is higher than that of VA with chlorine dioxide solution, although the consumption of VE is smaller than that of VA". The author needs further explanation in "Results and Discussion".

Review form: Reviewer 2 (Ying Guo)

Is the manuscript scientifically sound in its present form?

Yes

Are the interpretations and conclusions justified by the results?

Yes

Is the language acceptable?

Yes

Is it clear how to access all supporting data?

Yes

Do you have any ethical concerns with this paper?

No

Have you any concerns about statistical analyses in this paper?

No

Recommendation?

Accept with minor revision (please list in comments)

Comments to the Author(s)

The adsorbable organic halogen (AOX) mainly comes from the reaction of residual lignin and chlorinated bleaching agent during bleaching. The application of lignin model compound in pulping is helpful to understand the reaction mechanism of lignin and chlorine dioxide from the molecular level. The pathway to AOX formation by the lignin model compound and chlorine dioxide was speculated, which will provide a theoretical basis for reducing the formation of AOX. I believe the manuscript can be accepted after minor revision. Some specific points are showed below.

2. Introduction. "Many studies have focused on the degradation of lignin model compounds". What is the difference between this study and previous study.

3.2. Experiments. "The reaction mixture was stored in a refrigerator and analyzed as soon as possible". How to ensure the reaction did not continue.

3.4. HPLC Analysis. "using a mobile phase composed of acetonitrile-water (10/90) (vol) and 0.1% acetic acid, a flow rate of 0.3 mL min⁻¹, a temperature of 50 °C, and UV detection at 280 nm". If there are some references, please insert them.

4.1. Effect of reaction time on AOX formation. "Each individual reaction was performed in a solution with an initial AOX content of 6.2 mmol and concentration of 124 mmol ·L⁻¹". Please verify if "AOX content" should be "lignin model compound".

4.1. Effect of reaction time on AOX formation. "The AOX content increased within 10 minutes and ultimately reached 1224 mg ·L⁻¹". Please revise the time unit of the manuscript.

4.2. Concentration of lignin model compounds. "The AOX content formed by the reaction of VE with chlorine dioxide solution was 60.9 mg ·L⁻¹ creating a total concentration of 12.4 mmol ·L⁻¹, which is slightly higher than that formed by VA with chlorine dioxide solution (55.4 mg ·L⁻¹)". The author needs further explanation in this result.

4.2. Concentration of lignin model compounds. "The total AOX content of the separate reaction is less than that of combined reaction of VA and VE with chlorine dioxide solution at the same total concentration". The author needs further explanation in this result.

Decision letter (RSOS-182024.R0)

18-Dec-2018

Dear Dr Yao:

Title: Effect of lignin structure on AOX formation in chlorine dioxide bleaching
Manuscript ID: RSOS-182024

The editor assigned to your manuscript has now received comments from reviewers. We would like you to revise your paper in accordance with the referee and Subject Editor suggestions which can be found below (not including confidential reports to the Editor). Please note this decision does not guarantee eventual acceptance.

Please submit your revised paper before 10-Jan-2019. Please note that the revision deadline will expire at 00.00am on this date. If we do not hear from you within this time then it will be assumed that the paper has been withdrawn. In exceptional circumstances, extensions may be possible if agreed with the Editorial Office in advance. We do not allow multiple rounds of revision so we urge you to make every effort to fully address all of the comments at this stage. If deemed necessary by the Editors, your manuscript will be sent back to one or more of the original reviewers for assessment. If the original reviewers are not available we may invite new reviewers.

Please also include the following statements alongside the other end statements. As we cannot publish your manuscript without these end statements included, if you feel that a given heading is not relevant to your paper, please nevertheless include the heading and explicitly state that it is not relevant to your work.

- Ethics statement

Please clarify whether you received ethical approval from a local ethics committee to carry out your study. If so please include details of this, including the name of the committee that gave consent in a Research Ethics section after your main text. Please also clarify whether you received informed consent for the participants to participate in the study and state this in your Research Ethics section.

OR

Please clarify whether you obtained the necessary licences and approvals from your institutional animal ethics committee before conducting your research. Please provide details of these licences and approvals in an Animal Ethics section after your main text.

OR

Please clarify whether you obtained the appropriate permissions and licences to conduct the fieldwork detailed in your study. Please provide details of these in your methods section.

RSC Associate Editor:
Comments to the Author:
(There are no comments.)

RSC Subject Editor:
Comments to the Author:
(There are no comments.)

Reviewers' Comments to Author:
Reviewer: 1

Comments to the Author(s)

Effect of lignin structure on adsorbable organic halogens (AOX) formation was investigated by authors. Vanillyl alcohol (VA) and veratryl alcohol (VE) were selected for the phenolic and non-phenolic lignin model compounds, respectively. It has practical and scientific significance in elucidating the reaction pathway of lignin with ClO₂ and therefore reducing chlorinated pollutants. I believe the manuscript can be accepted after the following minor revisions.

2. Introduction. "The chemical structure of lignin is complex, and the lignin macromolecules have a great variety of bonds". Why were phenolic lignin model compound and non-phenolic lignin model compound selected to study?

3.2. Experiments. "Then, the respective lignin model compound was transferred to the chlorine dioxide solution and reacted at 70 °C with magnetic stirring". How many rpm? Please explain the experiment more clearly.

4.1. Effect of reaction time on AOX formation. "This is in agreement with the conclusions of C. Bogatu, who found that the oxidation of phenolic compounds with chlorine dioxide in water solutions contains a very fast step". Please insert references.

4.1. Effect of reaction time on AOX formation. "AOX was also formed rapidly at the beginning of

the reaction when VE reacted with chlorine dioxide, reaching 1036 mg•L⁻¹ after 10 minutes". Please revise the time unit of the manuscript.

5. Conclusion. "The AOX content produced by the reaction of VE with chlorine dioxide solution is higher than that of VA with chlorine dioxide solution, although the consumption of VE is smaller than that of VA". The author needs further explanation in "Results and Discussion".

Reviewer: 2

Comments to the Author(s)

The adsorbable organic halogen (AOX) mainly comes from the reaction of residual lignin and chlorinated bleaching agent during bleaching. The application of lignin model compound in pulping is helpful to understand the reaction mechanism of lignin and chlorine dioxide from the molecular level. The pathway to AOX formation by the lignin model compound and chlorine dioxide was speculated, which will provide a theoretical basis for reducing the formation of AOX. I believe the manuscript can be accepted after minor revision. Some specific points are showed below.

2. Introduction. "Many studies have focused on the degradation of lignin model compounds". What is the difference between this study and previous study.

3.2. Experiments. "The reaction mixture was stored in a refrigerator and analyzed as soon as possible". How to ensure the reaction did not continue.

3.4. HPLC Analysis. "using a mobile phase composed of acetonitrile-water (10/90) (vol) and 0.1% acetic acid, a flow rate of 0.3 mL min⁻¹, a temperature of 50 °C, and UV detection at 280 nm". If there are some references, please insert them.

4.1. Effect of reaction time on AOX formation. "Each individual reaction was performed in a solution with an initial AOX content of 6.2 mmol and concentration of 124 mmol L⁻¹". Please verify if "AOX content" should be "lignin model compound".

4.1. Effect of reaction time on AOX formation. "The AOX content increased within 10 minutes and ultimately reached 1224 mg L⁻¹". Please revise the time unit of the manuscript.

4.2. Concentration of lignin model compounds. "The AOX content formed by the reaction of VE with chlorine dioxide solution was 60.9 mg L⁻¹ creating a total concentration of 12.4 mmol L⁻¹, which is slightly higher than that formed by VA with chlorine dioxide solution (55.4 mg L⁻¹)". The author needs further explanation in this result.

4.2. Concentration of lignin model compounds. "The total AOX content of the separate reaction is less than that of combined reaction of VA and VE with chlorine dioxide solution at the same total concentration". The author needs further explanation in this result.

Author's Response to Decision Letter for (RSOS-182024.R0)

See Appendix A.

Decision letter (RSOS-182024.R1)

14-Jan-2019

Dear Dr Yao:

Title: Effect of lignin structure on AOX formation in chlorine dioxide bleaching
Manuscript ID: RSOS-182024.R1

It is a pleasure to accept your manuscript in its current form for publication in Royal Society Open Science. The chemistry content of Royal Society Open Science is published in collaboration with the Royal Society of Chemistry.

RSC Associate Editor
Comments to the Author:
(There are no comments.)

Reviewer(s)' Comments to Author:

Appendix A

Dear Reviewer,

Thank you for your letter and for the comments concerning our manuscript entitled “Effect of lignin structure on AOX formation in chlorine dioxide bleaching”. We have studied your comments carefully and have made corrections which we hope could meet your requirements. All changes have been highlighted in the revised version (red highlighting).

Questions you put forward are explained as follows:

Reviewer #1:

2. Introduction. “The chemical structure of lignin is complex, and the lignin macromolecules have a great variety of bonds”. Why were phenolic lignin model compound and non-phenolic lignin model compound selected to study?

Phenolic hydroxyl group is an important functional group of lignin structure. Lignin can be divided into phenolic lignin and non-phenolic lignin in term of phenolic hydroxyl group. The phenolic hydroxyl groups can influence the main types of reaction and the reactivity of lignin structures in chemical reaction. Therefore, the influence of phenolic and non-phenolic lignin model compounds on AOX formation was studied.

3.2. Experiments. “Then, the respective lignin model compound was transferred to the chlorine dioxide solution and reacted at 70 °C with magnetic stirring”. How many rpm? Please explain the experiment more clearly.

The rpm of magnetic stirring was 150. It has been described more clearly in the revised manuscript.

4.1. Effect of reaction time on AOX formation. “This is in

agreement with the conclusions of C. Bogatu, who found that the oxidation of phenolic compounds with chlorine dioxide in water solutions contains a very fast step”. Please insert references.

The reference was added to the revised version.

4.1. Effect of reaction time on AOX formation. “AOX was also formed rapidly at the beginning of the reaction when VE reacted with chlorine dioxide, reaching 1036 mg•L⁻¹ after 10 minutes”. Please revise the time unit of the manuscript.

The units of time have been modified in the revised version.

5. Conclusion. “The AOX content produced by the reaction of VE with chlorine dioxide solution is higher than that of VA with chlorine dioxide solution, although the consumption of VE is smaller than that of VA”. The author needs further explanation in “Results and Discussion”.

Under the condition of excessive lignin model, less VE was consumed by quantitative chlorine dioxide. This is due to the low reactivity of non-phenolic lignin. The complete reaction of VE requires more chlorine dioxide than VA. VA containing phenolic structure has higher reactivity. Phenolic hydroxyl groups were converted into free radicals for it was easily attacked by chlorine dioxide. And then the oxidation fracture of the benzene ring. Most free radicals are further oxidized by chlorine dioxide to form chlorolipids. It was further decomposed into muconic acid, ortho-quinone, para-quinone and oxacyclopropane structures for its unstable structure. Oxidative fracture of aromatic structures was the main reaction of VA in chlorine dioxide solution. Therefore, the lower content of AOX was generated from VA. VE has lower

reactivity due to the absence of phenolic hydroxyl group. The hydroxymethyl side chain was easier oxidized by chlorine dioxide. Then chlorine substitution was occurred on the aromatic-ring. More content of AOX was generated at a lower reaction rate. It has been explained more clearly in the revised version.

Reviewer #2:

2. Introduction. “Many studies have focused on the degradation of lignin model compounds”. What is the difference between this study and previous study.

Previous study mainly focused on the changes of structure and degradation pathway of lignin model compounds with specific structures. In this study, the effect of lignin structures on AOX formation and types of chlorinated products by different structures was studied.

3.2. Experiments. “The reaction mixture was stored in a refrigerator and analyzed as soon as possible”. How to ensure the reaction did not continue.

The reaction was terminated by the oxidation of sodium sulfite with residual chlorine dioxide. In addition, the reaction mixture was diluted immediately in deionized water to achieve reaction termination. It has been described more clearly in the revised version.

3.4. HPLC Analysis. “using a mobile phase composed of acetonitrile-water (10/90) (vol) and 0.1% acetic acid, a flow rate of 0.3 mL·min⁻¹, a temperature of 50 °C, and UV detection at 280 nm”. If there are some references, please insert them.

The reference has been inserted in the revised version.

4.1. Effect of reaction time on AOX formation. “Each individual reaction was performed in a solution with an initial AOX content of 6.2 mmol and concentration of 124 mmol·L⁻¹”. Please verify if “AOX content” should be “lignin model compound”.

It has been modified as required.

4.1. Effect of reaction time on AOX formation. “The AOX content increased within 10 minutes and ultimately reached 1224 mg·L⁻¹”. Please revise the time unit of the manuscript.

The units of time have been modified in the revised version.

4.2. Concentration of lignin model compounds. “The AOX content formed by the reaction of VE with chlorine dioxide solution was 60.9 mg·L⁻¹ creating a total concentration of 12.4 mmol·L⁻¹, which is slightly higher than that formed by VA with chlorine dioxide solution (55.4 mg·L⁻¹)”. The author needs further explanation in this result.

VA containing phenolic structure has higher reactivity. Phenolic hydroxyl groups were converted into free radicals for it was easily attacked by chlorine dioxide. And then the oxidative fracture of the benzene ring. Most free radicals are further oxidized by chlorine dioxide to form chlorolipids. It was further decomposed into muconic acid, ortho-quinone, para-quinone and oxacyclopropane structures for its unstable structure. Oxidative fracture of aromatic structures was the main reaction of VA in chlorine dioxide solution. Therefore, the lower content of AOX was generated from VA. VE has lower reactivity due to the absence of phenolic hydroxyl group. The hydroxymethyl side chain was easier oxidized by chlorine dioxide. Then chlorine substitution was occurred on the aromatic-ring. More content of AOX was generated at a lower

reaction rate.

4.2. Concentration of lignin model compounds. “The total AOX content of the separate reaction is less than that of combined reaction of VA and VE with chlorine dioxide solution at the same total concentration”. The author needs further explanation in this result.

Under the condition of excessive lignin model, less VE was consumed by quantitative chlorine dioxide. This is due to the low reactivity of non-phenolic lignin. The complete reaction of VE requires more chlorine dioxide than VA. In combination reaction of VA and VE, the dosage of chlorine dioxide reacting with VE increased. The consumption of VE increased comparing to that in separated reaction of VE. Meanwhile, the VA shared similar consumption in separated and combined reaction. Therefore, the total AOX content of the separate reaction is less than that of combined reaction at the same total concentration.

As a whole, issues the reviewer suggested are very pertinent, which are very helpful to modified my entire paper and thank you very much again.